# Effects of B-Cell Lymphoma on the Immune System and Immune Recovery after Treatment: The Paradigm of Targeted Therapy

**DOI:** 10.3390/ijms23063368

**Published:** 2022-03-21

**Authors:** Salvatrice Mancuso, Marta Mattana, Melania Carlisi, Marco Santoro, Sergio Siragusa

**Affiliations:** 1Department of Health Promotion, Mother and Child Care, Internal Medicine and Medical Specialties, University of Palermo, 90127 Palermo, Italy; mattana_marta@yahoo.it (M.M.); marco.santoro03@unipa.it (M.S.); sergio.siragusa@unipa.it (S.S.); 2Hematology Unit, Department of Oncology, AOUP “P. Giaccone”, 90127 Palermo, Italy; melaniacarlisi@yahoo.it

**Keywords:** B-cell lymphoma, immunoevasion, immunosuppression, immunosenescence, chemotherapy, immune recovery, targeted therapy, immune therapy, CAR-T

## Abstract

B-cell lymphoma and lymphoproliferative diseases represent a heterogeneous and complex group of neoplasms that are accompanied by a broad range of immune regulatory disorder phenotypes. Clinical features of autoimmunity, hyperinflammation, immunodeficiency and infection can variously dominate, depending on the immune pathway most involved. Immunological imbalance can play a role in lymphomagenesis, also supporting the progression of the disease, while on the other hand, lymphoma acts on the immune system to weaken immunosurveillance and facilitate immunoevasion. Therefore, the modulation of immunity can have a profound effect on disease progression or resolution, which makes the immune system a critical target for new therapies. In the current therapeutic scenario enriched by chemo-free regimens, it is important to establish the effect of various drugs on the disease, as well as on the restoration of immune functions. In fact, treatment of B-cell lymphoma with passive immunotherapy that targets tumor cells or targets the tumor microenvironment, together with adoptive immunotherapy, is becoming more frequent. The aim of this review is to report relevant data on the evolution of the immune system during and after treatment with targeted therapy of B-cell lymphomas.

## 1. Introduction

B-cell lymphoma represents one of the most active fields of clinical and biological research at the present time. The growing body of molecular discoveries has successfully supported the development of informed therapeutic strategies [1,2]. Most patients with indolent or aggressive non-Hodgkin lymphoma (NHL) can be cured with initial chemoimmunotherapy [3]. For patients with relapsed disease, a number of therapies are currently available or under investigation, ranging from drugs that target multiple pathways to adoptive cellular therapies that harness the patient’s immune system to fight the disease [4]. In this complex scenario, there are many challenges that we face when treating patients with lymphoma. Beyond the multiple considerations related to disease characteristics and the efficacy of different regimens that must be taken into account when selecting a treatment, additional work is required to unravel the interactions between the immune system, lymphoma and therapies. The successful introduction of targeted therapies is serving as a strong accelerator for the recognition of the complexity, diversity and clinical relevance of the role of the immune system throughout all stages, from lymphomagenesis to survival after cure. In this paper, we perform a review of the data supporting the effects of different B-cell lymphomas on immune functions. Another related topic covered here is the role of current, standard B-cell lymphoma treatments involving chemo, immunochemotherapy or both on the immune system. Furthermore, we examine available data on major novel developments in lymphoma therapy, paying particular attention to potential treatment-related immunological disorders. In order to achieve a more refined and dynamic picture, we focus on the impact of novel therapeutic approaches on the kinetics of immune recovery over time, which ultimately influences outcomes. Finally, we conclude with a discussion on the challenges and future directions of current immunological studies and how these can be integrated in the monitoring of therapy effectiveness.

An understanding of the complex immunological alterations produced by B-cell lymphoma and its treatments will more rationally orient therapeutic choice to improve survival by reducing or eliminating the risks of inflammatory and infectious adverse effects. On the other hand, the recognition of immune recovery can further help providers select drugs based on immunological endpoints.

## 2. Effects of B-Cell Lymphoma/Lymphoproliferative Diseases on Immune Functions

### 2.1. Chronic Lymphocytic Leukemia

Chronic lymphocytic leukemia (CLL) is a disorder of morphologically mature but immunologically incompetent B lymphocytes that accounts for about 25% of all leukemias [5]. Incidence increases with age, and the majority of patients are elderly, a circumstance that predisposes patients to a higher risk of infections [6]. Deficiencies in multiple arms of the immune system have been identified, further increasing the disorders related to immunosuppression [7]. Dysfunction of the immune system in CLL is very complex and is articulated in a variable way during the course of the disease. In the early stages, before treatment, there are already signs of immunological dysregulation that are at the origin of autoimmune phenomena. This suggests that moderate immune suppression can impair immune regulatory control of autoimmune responses [8]. Marked dysfunction of the innate immune response is seen in patients with CLL from the time of diagnosis [9].

In almost 40% of cases, reduced levels of some complement components are observed, particularly of C1–C4 [10]. C3b activity is also impaired, and these alterations together can contribute to the increased risk of infection and the reduced therapeutic action of immunoglobulins [11]. The mechanism that determines complement deficiency is unknown. A genetic origin has been hypothesized, considering the finding of complement deficiency in healthy family members of patients with CLL.

A number of qualitative defects in neutrophils have been observed, including an impaired phagocytic killing of nonopsonized bacteria and a reduction in chemotaxis [12]. As for blood circulating monocytes, these are often observed in higher numbers [13] and present a nonclassical CD14+ CD16+ phenotype and a gene-expression profile associated with immunosuppressive activity. It appears that the immunosuppressive profile derives from a direct action of the CLL clone [14]. In a similar way, Natural Killer (NK) cells are numerically increased but functionally ineffective, with impairment of cytotoxic activity caused by the downregulation of the NKG2D receptor, an activating receptor that plays a key role in the immune response against cancer [15]. CLL is characterized by a deep and complex alteration on the T-cell compartment. The distribution of early T-lymphocyte populations is abnormal, with reduced T naïve and stem cell memory (TSCM) cells and elevated terminally differentiated cells (T terminal effector memory, T effector memory and CD8+ terminal effector memory cells) [16,17]. The altered distribution of T lymphocytes in the different stages correlates with impairment of immune response [18,19].

Furthermore, effector and memory T cells in CLL patients are frequently pseudoexhausted with increased anergy due to chronic activation by CLL B cells and interactions with other immunosuppressive cells, such as regulatory T cells (Tregs) and myeloid-derived suppressor cells (MDSCs) These pseudoexhausted T cells may be a result of chronic stimulation by low-affinity self-antigens [20,21].

Elevated circulating Tregs have been observed in CLL with a negative prognostic impact [22]. MDSCs also increase during the course of CLL and further enhance the pressure on T lymphocytes with immunosuppressive effects [23]. Following T-Cell receptor (TCR) stimulation, T lymphocytes in CLL have high rates of apoptosis as a result of chronic activation. T-cell potential proliferative ability is also impaired in CLL patients, leading to additional immunosuppressive action [16,24]. Another key characteristic of immune dysfunction in CLL is impaired T-cell effector function. The normal release of cytokines and lytic proteins by T lymphocytes upon antigen recognition and costimulatory signals is reduced. The impaired secretion of interleukin (IL)-4, IL-6, IL-10 and IL-17A was evidenced, especially in patients with poor prognostic features [16]. The profound immune dysregulation that distinguishes CLL, as well as establishing a high infectious risk, acts in determining the progression of the disease. Malignant B cells trigger immunoevasion phenomena, co-opting MDSCs and Tregs that accumulate in the tumor microenvironment. These cells, in turn, promote the growth and survival of the malignant B cells [25,26]. The main immunological feature of the adaptive immune response in CLL is represented by hypogammaglobulinemia, which significantly correlates to an increased risk of common bacterial infections [27].

Reduced immunoglobulin (Ig) levels can already be observed in the early stages of the disease and become more severe as CLL progresses [28]. All Ig classes are involved, with the depletion being greatest in the IgG3 and IgG4 subclasses [7,29]. The efficiency of vaccine response is also compromised from the earliest stages of the disease. This significant clinical aspect is due to the combined effect of the direct suppression of B-cell function and the impairment of T-cell helper function [30,31].

In light of these complex and profound alterations transversally involving almost all the players of the immune response, the mechanisms behind these changes remain to be identified. Based on the available data, the most likely scenario is represented by the immunosuppressive role directly played by neoplastic B cells. The ability of CLL cells to inhibit the functions of normal T and B lymphocytes by direct cell contact and tumor-released soluble factors has been demonstrated. The immunoregulatory function of CLL cells appears to be supported by a large battery of molecules, whose mechanisms of action have been described. The proposed model refers to a similar profile between the neoplastic B cells and the recently described regulatory B (Breg) cells. These cells represent a fraction of B cells, equal to 4%, and perform regulatory functions mainly through a soluble factor, IL-10. They expand during inflammation, autoimmunity and cancer, and carry out inhibition and regulation activities on innate and adaptive immunity. CLL cells share multiple phenotypic markers with IL-10-competent regulatory B cells and can secrete IL-10 following appropriate stimulation [32,33]. It should be emphasized that increased levels of IL-10 have been associated with a diminished survival in CLL patients.

All of these alterations are of enormous relevance in light of the ongoing COVID-19 pandemic, since patients with CLL may be at high risk of poor outcomes if they are hospitalized due to COVID-19 [34,35,36].

Until a few years ago, this complex immunopathological scenario was further compromised by the impact of immunochemotherapy. Alkylating drugs, such as bendamustine, chlorambucil or cyclophosphamide, and purine analogs, such as fludarabine, pentostatin or cladribine, combined with anti-CD20 monoclonal antibodies, although effective in controlling the disease, have heavy side effects on the immune system. In fact, chemotherapy also reduces healthy dividing cells, including T lymphocytes, slowing down recovery times [37,38]. After treatment, during follow-up, infectious events tend to accumulate, and immunological impairment becomes irreversible. Therefore, CLL patients treated according to previous-generation therapies remain a population at high risk of morbidity and mortality. From this evidence, the therapeutic approach and treatment objectives were revised.

### 2.2. Indolent Lymphoma

Within the broad category of B-cell lymphomas, on the basis of their histopathological features and according to the World Health Organization (WHO) classification, we include a number of disease entities with an often-chronic clinical course, mainly affecting the elderly. Like CLL, these conditions arise in the adaptive immune response arm. The natural history of these indolent entities is characterized by a relatively slow-growing with a low potential for cure and the median survival measured in years to decades. From a pathogenetic point of view, mutual relationships between the immune system and the lymphoma are established. Indeed, immune activation mediated by infectious agents or autoantigens may promote the development and the progression of the lymphoma. On the other hand, B-cell indolent lymphoma has a complex action on the immune system that is predominantly inhibitory. These effects result in a clinical condition of immunodeficiency. In this context, the role played by immunosenescence on lymphomagenesis should be considered in view of the peculiar association of some indolent lymphomas with aging. In addition, current and emerging therapies intercept the specific immunological context, inserting additional elements of immune dysfunction and dysregulation. As a result, clinical frameworks strongly conditioned by the final alterations of the immune response may develop.

At the basis of the progress of several new immunological therapies lies the expansion of knowledge on the molecular mechanisms of the tumor microenvironment that enable the onset and progression of lymphoma. The normal antitumor immunity performed by CD8+ T cells is functionally incompetent in lymphoma tissue. The mechanisms that induce this inhibition are different. Among them, the roles of Treg cells [39], inhibitory cytokines [40], T-cell exhaustion [41,42] and terminal differentiation [43] have been demonstrated in follicular lymphoma as well as other histotypes. A study carried out on distinct CD8+ subsets in follicular lymphoma (FL) is of significant interest for potential clinical implications [44]. Intratumoral KLRG + CD127- and KLRG-CD127- CD8+ effector cells prevail over memory cell subsets characterized by a KLRG + CD127+ or KLRG-CD127+ profile. The proliferative capacity of the effector cells is inferior when compared with memory subsets; this imbalance has prognostic significance as it relates to inferior event-free survival (EFS) and overall survival (OS). The expansion of KLRG + CD127- cells appears to be promoted by interleukin (IL)-15 in the presence of dendritic cells via a phosphoinositide 3-kinase (PI3K)- dependent mechanism, and the inhibition of PI3K in CD8+ cells seems to downregulate the differentiation of effector-cell subsets. As in other observations, understanding the biology of immune infiltration in patients with FL at risk of early progression is a promising tool [45]. Another study showed that peripheral T-cell subsets in untreated FL patients differ from those in healthy donors, with a higher proportion of activated T cells expressing programmed cell death protein-1 (PD-1)/T-cell immunoreceptor with Ig and ITIM domains (TIGIT). Patients with lower percentages of naïve CD4+ cells at baseline exhibit a worse response to rituximab-based therapy. These data provide the possible perspective that the pattern of T-lymphocyte subsets may acquire a prognostic significance in FL [46].

A prospective controlled study in follicular lymphoma and extranodal marginal-zone lymphomas demonstrates reduced circulating T helper cells, particularly naïve CD4+ cells. In addition, two patterns of T-helper dysfunction were observed: first, a chronic immune activation with T helper 2 cells (Th2) shift, in vitro hyperreactivity and T-cell senescence with a propensity to undergo apoptosis; and second, a downregulation of TCR signaling and cytokine secretion [47]. The clinical consequences of impaired immunity in indolent lymphomas can be quite relevant. It is interesting that anatomically localized action between B-lymphoma cells and T-lymphocytes can produce systemic immune dysfunction. These data provide a knowledge base to understand the increased susceptibility to infection and the reduced immune response to vaccines in patients with indolent B-cell lymphoma. In addition, the molecular link represented by immune activation/Th2 dysregulation patterns clarifies some aspects of the causal role of autoimmunity in lymphomagenesis.

Immunochemotherapy adds further cause of immune system impairment. Patients with previously untreated CLL and indolent lymphoma who received bendamustine with rituximab (BR) showed impairment of CD4+ T-cell and CD8+ T-cell count recovery six months after treatment finished. The same dynamic was observed for CD19+ cells and the NK-cell compartment. In comparison to rituximab with –cyclophosphamide, doxorubicin, vincristine and prednisolone (R-CHOP), BR treatment was found to be more immunosuppressive [48].

### 2.3. Aggressive B-Cell Lymphoma

#### 2.3.1. Diffuse Large B-Cell Lymphoma

Diffuse large B-Cell lymphomas (DLBCL) account for 30–58% of all diagnosed non-Hodgkin lymphomas. DLBCL is a constellation of heterogeneous entities with a high degree of malignancy, each with distinct morphologic, biologic and clinical characteristics. Due to this heterogeneity, nearly 40% of patients do not fully benefit from the existing immunochemotherapeutic approach. The prognosis appears to correlate with the tumor microenvironment, its cellular composition, infiltrating immune cells and gene expression affecting a variety of infiltrating immune cells [2]. However, to date, only a few patients have benefitted from a programmed death ligand-1 (PD-L1) and PD-1 blockade, suggesting the need to stratify patients on the basis of the immunological and genetic profile linked to the organization of the microenvironment [49].

The links between DLBCL and immune system alterations are numerous and complex. Certain autoimmune diseases, immune deficiency syndromes and infections may contribute to the pathogenesis of DLBCL. On the other hand, this type of cancer is known to result in immunological dysfunction, reducing normal B cells and the normal communication between B cells and other immune cells [50]. Moreover, DLBCL treatments, including chemotherapy, rituximab, corticosteroids and radiotherapy, all have profound effects on immune networks. 

A special setting is represented by the DLBCL in the elderly, which constitutes the prevailing group [51]. In this subset, the lymphoma–host interactions generate numerous conditions that can affect immunological status and influence infectious risk and, consequently, morbidity and mortality [52,53]. These patients may present a collection of unfavorable features, such as a high International Prognostic Index (IPI) score and a high number of comorbidities. In addition, secondary infectious risk can be amplified by immunochemotherapy treatment, specifically R-CHOP. In a retrospective study evaluating DLBCL patients aged > 70 years who received a full or attenuated dose of R-CHOP, the predictive variables of increased risk of infection-related morbidity and mortality were the intended dose intensity > 80 of R-CHOP with an IPI score of 3 to 5, a Cumulative Illness Rating Scale-Geriatric (CIRS-G) score > 6, and low albumin [54]. In this cohort, 51 of 690 treated patients died as a result of infection. In another study, it was shown that increased infectious risk in the elderly persists up to five years after treatment [55]. In a large retrospective cohort study of 21,690 DLBCL survivors, elevated incidence rate ratios for immune-related events were found 5–10 years after cancer diagnosis. Significantly increased risks of viral and fungal pneumonia, meningitis, humoral deficiency and autoimmune cytopenias were observed. This deep and long-lasting impairment of immune health can only partially be explained by the effects of treatment, but also likely depend on the complex biology of DLBCL [56].

#### 2.3.2. Mantle Cell Lymphoma 

Mantle cell lymphoma (MCL) is a distinct subtype of B-cell non-Hodgkin lymphoma accounting for 6% of all lymphomas and primarily affects individuals at a median age of 65 years. MCL has a broad spectrum of clinical, biological and genetic features. The main conventional entity presents a more aggressive clinical course, with successive phases of response and relapses and a median survival of 3–6 years. The current first-line therapy includes intensive chemotherapy, containing, e.g., high doses of aracytin, with autologous stem-cell transplantation in younger, fit patients. Tumor cells express an antigenic pattern, which makes them susceptible to targeted therapies [57]. The effects of MCL on the immune system have not been extensively analyzed. In some studies, the expression of PL-1 on tumor cells has been described, and this could support the mechanism of immunoevasion and inhibition of the antitumor T-cell response [58,59]. Figure 1 shows the interrelationship between B-cell lymphoma and the immune system. An imbalance of the immune system is closely related to the onset and progression of B-cell lymphomas. In addition, the same lymphoma acts on the immune system to promote its progression. Finally, in synergy with immunochemotherapy, the lymphoma causes a further imbalance of immune defenses.

## 3. Effects of Targeted Therapy on Immune Functions

Targeted drugs have fundamentally changed the treatment landscape of B-cell malignancies.

Approaches to targeted and immunological therapies in lymphoproliferative diseases fall into the following categories: therapies that target immunological markers on the tumor cells themselves; therapies that target immune effector cells and enhance or direct an antineoplastic effect; and adoptive T/NK cell therapy.

An increased awareness about the possible risk of infection with this heterogeneous category of drugs has produced detailed reports on the infectious emergency related to new treatments [60]. However, the effects of these new treatments on innate and adaptive immunity are not fully elucidated yet. The biological impact caused by different agents on the immune system will provide the rationale for monitoring infectious complications.

### 3.1. Agents Targeting Hematological Cells

#### 3.1.1. Anti-CD20 Direct Agents: Rituximab and Obinutuzumab 

Rituximab is the first monoclonal antibody targeting CD20 used for the treatment of a multitude of B-cell malignancies, allowing significant successes to be achieved with passive immunotherapy. CD20 is a cell surface marker unique to B cells. As a consequence of the main action performed on the CD20 target, the effect of rituximab on the immune side has been shown to be reduced serum immunoglobulin levels [61]. A further long-term effect on immune defenses is a late-onset reduced neutrophil count, the origin of which appears to be immune-mediated. This peculiar condition arises 1–5 months after the end of therapy in 5–15% of subjects treated with rituximab [62].

The main aspect to be clarified is the effect of rituximab on B–T-cell interactions and cellular immune-impairment-related infections. Established evidence demonstrates a more-than-five-fold increase in human hepatitis B virus (HBV) reactivation, so that screening for latent infection is recommended [63]. The role of rituximab in increasing Pneumocystis pneumonia (PCP) cases in patients with B-cell lymphoma treated with immunochemotherapy is not clear. In the elderly, a randomized trial comparing a CHOP regimen (cyclophosphamide, hydroxydaunorubicin, oncovin, prednisone) with R-CHOP (rituximab plus CHOP) did not report cases of PCP [64]. However, a greater risk of PCP has been described in patients treated with biweekly R-CHOP [65,66] and in solid transplant recipients receiving rituximab for the treatment of rejection [67]. Finally, a meta-analysis demonstrated a significantly increased PCP risk in lymphoma patients subjected to rituximab during chemotherapy [68]. Unlike human immunodeficiency virus (HIV) subjects, the immunological background and the role of CD4+ lymphocytes that would favor the onset of PCP in HIV-negative lymphoma patients is not even known [69]. Once again, it is likely that the combined action of an anti-CD20 direct agent, together with steroids and chemotherapy, may produce immunosuppressive effects that contribute to opportunistic infections. In light of the effects of the COVID-19 pandemic in immunosuppressed subjects, a study in patients already treated with rituximab for lymphoma or autoimmune diseases showed an impaired humoral and cellular response to mRNA vaccines against SARS-Cov-2 [70]. A critical issue is the effect of single-agent rituximab on immune system restoration. After a significant depletion of B cells, recovery requires approximately one year following therapy [71]. The immunological studies have not shown significant effects on CD4+ or CD8+ T cells and NK cells [72]. 

In addition, there are few data on the dynamics of immune recovery after immunochemotherapy. In patients with B-cell lymphoma treated with CHOP-based chemotherapy containing rituximab, CD19+/CD20+, CD4+, CD3+, CD8+ and CD56+ cells underwent severe depletion during treatment. B cells and T cells required, respectively, one year and two years after therapy to recover to their levels at diagnosis. As regards Ig, their levels decreased during six cycles of therapy and recovered two years after therapy [73]. The effects on T cells are likely secondary to the CHOP regimen rather than to 3rituximab. In another study, CD4+ cell-count restoration after R-CHOP was very slow and not complete after 24 months. In contrast, B cells recovered to baseline levels in one year [74].

The effects of rituximab-based immunochemotherapy on T-cell functions are quite complex. In FL, rituximab-based therapy induces profound changes in peripheral T-cell subsets, with a shift from a T effector memory phenotype toward a T central memory phenotype for both CD4+ and CD8+ T-cell compartments, and also toward a naive phenotype for CD8+ T cells. Furthermore, a reduction in peripheral T cells expressing PD-1, TIGIT or both PD-1 and TIGIT immune checkpoints was highlighted [46]. It remains to be determined whether these changes have a long-term impact on the clinical outcome of FL patients. A study explored the ontogeny and characteristics of the reconstituting B-cell pool in patients with follicular lymphoma treated with rituximab. The majority of newly formed circulating B cells were phenotypically immature. Moreover, CD27+ memory B cells showed a significant delay in recovery. From these findings, it emerges that an immature immune system is produced ex novo, with numerous implications for maintenance schemes with rituximab and the consequent infectious risk [75]. Obinutuzumab, a type-II humanized anti-CD20 antibody, has shown superior antibody-dependent cellular cytotoxicity (ADCC) and direct cell death activity, and similar antibody-dependent cellular phagocytosis (ADCP) but inferior complement-dependent cytotoxicity (CDC) in vitro, versus rituximab. In CLL, at the immune system level, the main effects are observed on CD8+ cells and NK cells, with early and rapid depletion and subsequent recovery after six months [76].

#### 3.1.2. Other Antibodies

After rituximab, several other antibodies targeting B-cell markers have been evaluated in lymphoma treatment. Tafasitamab is an anti-CD19 monoclonal antibody that has shown activity as a single agent and in combination with lenalidomide in patients with relapsed or refractory B-cell lymphoma. In the phase-two study of L-MIND, this association demonstrated a high rate of complete response in patients with relapsed or refractory DLBCL who were ineligible for autologous stem-cell transplantation. In light of these results, the synergic action was partly attributed to the immunomodulatory effect carried out on peripheral NK cells, the median count of which increased at cycle eight compared to baseline [77]. Bispecific T-cell engager antibodies (BiTEs) are a further evolution in the field of monoclonal antibodies, designed to target both tumor antigens and T cells. Blinatumomab is a BiTE that transiently links CD3+ T cells to CD19+ B cells, inducing T-cell activation, T-cell proliferation and T-cell-mediated lysis of neoplastic cells. Approved for the treatment of Philadelphia-chromosome-negative relapsed or refractory B-precursor acute lymphoblastic leukemia (B-ALL), its potential therapeutic action has been explored in refractory or resistant DLBCL with a promising response rate (OR was 43%) [78]. In a phase-one study of B-NHL that included 28 patients with FL, a complete response was achieved in 40% of patients [79]. However, in contrast to B-ALL, the immune system impact of blinatumomab has not been reported for B-cell lymphoma [80,81]. One way to increase the efficacy of monoclonal antibodies has been to conjugate them to cytotoxic drugs. Loncastuximab tesirine is also an anti-CD19 antibody drug conjugate that contains a cytotoxic pyrrolobenzodiazepine dimer, which is released into the neoplastic cell after binding to CD19, with subsequent cell death. It has been studied in heavily pretreated DLBCL patients with 46–59% of ORR in completed clinical trials. In a study of relapsed or refractory FL and MCL treated with loncastuximab tesirine, 80% of patients had a response to therapy [82]. Long-term follow-up is required for these novel antibody-based treatments to determine the impact on overall survival and the immunological outcome [83].

### 3.2. Bruton’s Tyrosine Kinase Inhibitors: Ibrutinib, Acalabrutinib, Zanubrutinib

Ibrutinib, together with acalabrutinib and zanubrutinib, is an irreversible Bruton’s tyrosine kinase (BTK) inhibitor. BTK is a critical enzyme involved in the development, activation and survival of B lymphocytes [84,85]. Ibrutinib is the most widely used agent, for which more data are available on the effects produced on the immune system. By binding to BTK, ibrutinib intercepts the B-cell receptor signaling pathway, resulting in the inhibition of proliferation and survival of B tumor cells. Ibrutinib is approved for the treatment of a group of lymphoproliferative diseases: CLL, mantle cell lymphoma, Waldenstrom’s macroglobulinemia and marginal-zone lymphoma. The correlation between ibrutinib treatment and the emergence of infectious complications has been reported [86]. However, the increased susceptibility to infection seems to be due to a number of additional factors, such as previous lines of chemotherapies, neutropenia and concomitant steroid therapy. Finally, the emergence of infectious complications during ibrutinib therapy appears to be concentrated especially in the first 6–12 months of treatment [87,88,89]. In patients with CLL, cytomegalovirus end-organ disease has been anecdotally reported [90]. In a prospective study, plasma cytomegalovirus (CMV) DNAemia was detected in 30.4% of cases, but no patients developed either recurrent CMV DNAemia or CMV-related organ disease during the 180 days after starting ibrutinib. Patients were also found with increased numbers of functional CMV-specific T cells upon antigenic stimulation, thus explaining the rapid clearance of CMV DNA in peripheral blood [91]. Beyond the safety aspects, growing evidence suggests that ibrutinib influences diverse immunological functions in a complex way. It also binds to interleukin-2-inducible T-cell kinase (ITK), a component of the TCR pathway. By binding to BTK and ITK, ibrutinib can modulate B cells, T cells, MDSCs and mast cells, producing numerous effects on the immune response [17,24,92,93].

A study [94] demonstrated that ibrutinib can facilitate repopulation and functional recovery of adaptative immunity, both in previously untreated and relapsed or refractory CLL patients enrolled in RESONATE 2 and RESONATE trials. Compared with ofatumumab or chlorambucil, improvements have been observed in 21 immune blood cell subsets throughout the first year of therapy. Ibrutinib significantly decreased abnormal B-cell counts, regulatory T cells, effector/memory CD4+ and CD8+ T cells, exhausted and chronically activated T cells, NK cells, and MDSCs; maintained naive T cells and NK cells; and increased circulating monocytes. In addition, ibrutinib significantly restored T-cell proliferative ability, degranulation, and cytokine secretion. The effects on T cells are of primary importance, because their compartment is strongly compromised in CLL, and the restoration of T-cell functions correlates both with the clinical response and antileukemic action that reduces infectious risk. These observations are in agreement with what was reported in clinical settings and larger cohorts after a long period of follow-up, with decreased infection rates after six months of treatment with ibrutinib [95,96,97]. In patients treated with ibrutinib, a recovery of humoral immunity has been shown with an increase in Ig levels, probably due to the loss of the immunosuppressive effects of CLL clone on bone marrow plasma cells [98].

In a trial comparing treatment with ibrutinib-rituximab and chemoimmunotherapy with fludarabine, cyclophosphamide and rituximab (FCR), the ibrutinib-rituximab arm resulted in better progression-free survival and overall survival, with infectious complications of grade 3 or higher being less common (10.5% vs. 20,3%, *p* < 0.001) [99]. The already-known immunosuppressive action of FCR is therefore confirmed, proving the possibility of partial and progressive recovery of immune functions in CLL with targeted therapy. Moreover, ibrutinib has demonstrated the ability to restore immune competency, as it enhances the efficacy of chimeric antigen receptor T (CAR-T) cells in CLL, in which T cells undergo immune dysfunction [100]. There is also evidence that ibrutinib can help enhance the effectiveness of immune checkpoint inhibitors (ICI) through its action on ITK in CLL patients, restoring antitumor T-cell immune response [101]. These observations create an interesting rationale to combine different targeted and immunological therapies [102,103].

Ibrutinib is also approved for patients affected by mantle cell lymphoma after the first line of treatment. Considering the advanced median age and the immunochemotherapy already administrated, ibrutinib is indicated for patients with a potentially very compromised immune system. In the study published on the long-term follow-up (median 26.7 months) of the phase-two registration trial of ibrutinib in this setting, the most frequent infections were observed in the upper respiratory and urinary tracts. The majority of infections were self-limiting, not supported by opportunistic pathogens and, above all, reduced over time [104]. Therefore, despite the progressive immunological dysfunction in CLL and MCL from initial diagnosis to a relapsed or refractory state, aggravated by chemotherapy, ibrutinib can improve the immune response over time in both diseases (Table 1).

However, chronic BTK inhibition may be a concern for the role it plays in immune tolerance and the potential development over time of autoimmune phenomena. BTK is essential for human B-cell tolerance [104].

Nevertheless, currently available clinical data show a decreasing risk of treatment–emergent autoimmune cytopenias (AICs) and better control in CLL patients with pre-existing AICs treated with ibrutinib via a reduction in T helper 17 (Th17) cells [105,106,107].

Acalabrutinib is a selective inhibitor of BTK with indication for the treatment of CLL. Compared to ibrutinib, acalabrutinib has improved target specificity and, therefore, reduced toxicities. The effects on the immune system are multifaceted and partly conflicting. If on the one hand a reduction in immunosuppressive molecules, together with a decreased production of IL-10, are observed, the Th2-to-T helper 1 (Th1) switch in the CD4+ T-cell compartment is absent. In addition, acalabrutinib produces negative effects on macrophage and neutrophil functions with deleterious consequences on antimicrobial control [107]. In the largest phase-three acalabrutinib trial in the relapsing or resistance CLL setting, grade 3 infectious events occurred in 23% of patients [108,109].

Zanubrutinib, a BTK inhibitor sharing BTK selectivity with Acalabrutinib, leads to a reduction in PD-1 and cytotoxic T–lymphocyte antigen 4 (CTLA-4) expression and a decrease in Treg cells, but it has no effect on the TH1/TH2 ratio [110].

### 3.3. PI3K Inhibitors

PI3K is frequently overexpressed in B-cell lymphomas. It plays a major role in B-cell signaling as well as in Treg cells and MDSC function. 

The PI3K inhibitor group includes oral idelalisib, rigosertib and duvelisib. Idelalisib is approved for the treatment of chronic lymphocytic leukemia and relapsed or refractory FL. In CLL, idelalisib has shown an inhibitory effect on the proliferation, survival, adhesion and homing of tumor B cells. It also plays a role on the immunological side, reducing Treg cells and pro-tumor factors [111]. Although the studies of idelalisib in combination with other drugs were prematurely closed due to unacceptable safety profiles, as a single agent, idelalisib still remains a treatment option for FL [112]. Conversely, in this setting, BTK inhibitors are less compelling. To date, the effects of these agents on the immune system seem to be converging on immune dysfunction that causes pneumonitis, colitis and hepatitis, as well as an increased risk of opportunistic infections, when high-dose steroid treatment is added [113,114]. In regards to colitis, this can reflect impaired gut immunity and tolerance to commensal microbiota. An additional PI3K inhibitor, copanlisib, is intravenously and intermittently administered and, therefore, determines fewer immune adverse events [115]. Alongside the phenomena of immunological dysfunction shared by almost all drugs in this group, for duvelisib, a dual inhibitor of the delta and gamma isoforms of PIK3, the additional inhibition of the gamma isoform could inhibit T-cell and macrophage polarization, thus disrupting the tumor microenvironment believed to be important in the development and maintenance of indolent B-cell lymphomas [116]. 

### 3.4. Bcl-2 Homology 3 (BH3) Mimetics: Antiapoptotic Protein B-Cell Lymphoma 2 (BCL-2) Inhibitors

The most prominent function of BCL-2 protein is to regulate the initiation of the intrinsic pathway of apoptosis. Oral BCL-2 inhibitor venetoclax is one of most promising strategies to treat lymphoproliferative diseases, given the overexpression of BCL-2 protein in tumor cells. It represents an important treatment option in CLL as a single agent or in combination, in different settings, up front or in relapsed patients. Due to the pathognomonic translocation involving the BCL-2 gene on chromosome 18, and the immunoglobulin heavy-chain locus on chromosome 14 observed in FL, venetoclax was tested in this prototypical disease. In addition, other indolent lymphomas, such as lymphoplasmacytic lymphoma and marginal-zone lymphoma, have been included in trials testing the efficacy of this drug [117]. The responses were variable with respect to different diseases, but the drug continues to be tested in various combinations [118]. A potential immunosuppressive action of venetoclax is neutropenia. Venetoclax has been shown to suppress the production of neutrophils in vitro and in animal models [119]. Neutropenia is quite frequent in CLL trials, occurring in 40–50% of cases, with 15% of patients with 3–4 neutropenia experiencing severe infections [120]. These side effects seem to depend on numerous variables related to the clinical profile of patients and the characteristics of the underlying disease [121]. The effects of venetoclax on the T-cell compartment have been described, with the total number of CD4+ and CD8+ being reduced and naïve T cells being proportionally low. In addition, venetoclax reduces the number of immunosuppressive cells in the microenvironment and leads to an overproduction of inflammatory cytokines [122]. The number of NK cells is also transiently reduced. In CLL, venetoclax reduces the frequency of PD-1 CD8+ T cells [123]. There are no data on the clinical impact of these observations and the dynamics of recovery of immune functions. The combination of venetoclax plus anti-CD20 antibodies in clinical trials shows durable responses that are recorded even after the discontinuation of therapy, although, also in this setting, the effects on the immune system need to be further investigated [106]. In MCL, significant data are reported on the dynamics of the evolution of lymphocyte subsets in patients treated with ibrutinib in combination with venetoclax. In a relapsed setting, long-term treatment was associated with the rebalancing of immune subsets, especially of CD4+ and CD8+ effector and central memory T cells and NK cells. Furthermore, the normalization of T-cell cytokine production in response to TCR stimulation was observed [124]. Therefore, this chemo-free treatment provides the possibility of recovering immune functions that have been compromised by the lymphoma and previous treatments.

## 4. Agents Targeting Immune Effector Cells and Enhance or Direct Antineoplastic Effect

### 4.1. Immune Checkpoint Inhibitors

The ongoing investigation of drugs that enhance the host immune response against tumor cells are of particular interest in B-cell lymphomas, given the clear graft-versus-lymphoma effect seen after allogenic stem-cell transplant, demonstrating that these cancers are highly immunoresponsive [125]. At the moment, there is still a long way to go before a wider use of ICIs in B-NHL can be recommended. The two main inhibitory mechanisms described are CTLA-4 and PD-1 and PDL1/PD1-ligand 2 (PDL2). The pharmacological blockade of these pathways by monoclonal antibodies allowed the recovery of the antitumoral activity of the immune system, mainly in the setting of solid tumors [126]. However, the success of anti-PD1 therapy in Hodgkin’s lymphoma continues to support the possibility of using this category of drugs also on B-cell NHL. ICIs such as nivolumab, pembrolizumab and pidilizumab have been tested in follicular lymphoma, with results that are not definitive although encouraging [127,128,129]. Ongoing studies are evaluating a possible synergistic effect of the combination of PD-1- or PD-L1-blocking agents with chemotherapy, other targeted therapies or other immunotherapeutic approaches in B-NHL. The role of additional factors, such as lifestyle, metabolic disorders and sociological factors, in determining the response to this category of drugs must also be considered [130]. The antitumoral effects of ICIs can be associated with the risk of a flare of immune-mediated adverse events in multiple organs. The next step in this scenario is immunosuppression caused by the treatment of the adverse inflammatory reactions with glucocorticoids or other immunosuppressant agents. Information about the emergence of secondary infections derives only from studies conducted in patients with solid tumors [131]. Furthermore, there are no data on the timing and rebalancing patterns of the immune system in the lymphoma setting. However, the intriguing role of this therapeutic approach in blocking the mechanism of immunoevasion will allow future evaluations of the restoration of the immune response in parallel with a desirable, long-lasting cure for B-cell lymphoma.

### 4.2. Lenalidomide

Lenalidomide was the first immunomodulating drug that exploited the immune system to perform an antitumor action in CLL. Its targets and immunological effects are multiple, including upregulation of CD40 and reduced expression of PD-1 on neoplastic cells, T-cell activation, increased immune synapse and tumor lysis. Among the immunomodulatory effects, a decrease in the Tregs count and a rise in Ig production by normal polyclonal B cells have been described [15]. The cytotoxic activity of NK cells was diminished in CLL patients; however, treatments with IL-2, IL-15, IL-21 and lenalidomide were able to restore their activity. The effects of IL-2 and IL-15 were associated with an increase in NKG2D expression on immune cells, but the effects of IL-21 and lenalidomide were not due to NKG2D upregulation [132].

## 5. Adoptive T/NK Cell Therapy

### CAR-T Cell Therapy

CAR-T cell therapies have had a remarkable impact on the treatment and prognosis of relapsed and refractory aggressive B-NHL, with up to 40% of patients achieving durable remission [133,134]. The efficacy is associated with a significant risk of cytokine release syndrome (CRS) and neurologic toxicity, which has, to some degree, limited the extent of its use for indolent lymphoma [135]. CAR-T-cell-associated toxicities are related to severe, supraphysiologic cytokine production and massive in vivo T-cell expansion. However, the latest studies are verifying the efficacy and safety results of this therapy in larger cohorts of patients with indolent B-NHL [136]. In addition, we are waiting to understand how CAR-T cells may impact the immunological landscape of this disease alongside its beneficial effect on the tumor cells. At the moment, clinical data on infectious complications are available, which may be indicators of the immunological state produced by treatment [137,138,139]. A significant risk of serious bacterial infections in the first 30 days after infusion has been reported, with more than a third of patients affected [140]. This high frequency correlates with previous antibacterial treatment and multiple cycles of chemotherapy responsible for the destruction of the composition of the microbiome and the emergence of multi-drug-resistant germs. Viral infections are more frequent later and can be severe. Fungal infections and CMV reactivation are less frequent [141]. Patients treated with CAR-T cells have a significant burden of infectious risk sustained by a unique immunobiology with a high degree of multifactorial immunosuppression. Prior chemotherapies (including cytotoxic, autologous or allogenic hematopoietic stem-cell transplant and lymphocyte-depleting chemotherapy) contribute to marrow suppression and the resulting cytopenias [142]. Neutropenia and lymphocytopenia are fairly universal before and after infusion. The duration of severe neutropenia can also be prolonged in 16% of cases, maintaining a high risk of infection over time [143]. In a retrospective study including 19 patients who received CD19 CAR-T cells, the absolute count of neutrophils and monocytes compared to the absolute lymphocyte count predicted infectious complications [144]. CAR-T cells targeting the B-cell lineage hold the potential for inducing B-cell depletion, long-term immune system dysfunction and hypogammaglobulinemia. A 15% incidence of hypogammaglobulinemia in DLBCL after treatment with CAR-T cells has been reported. Hypogammaglobulinemia can contribute to viral infections in this population, compromising the production of viral-specific neutralizing antibodies [145]. On the contrary, few data are reported on immunological recovery. In a study, patients with DLBCL treated with CAR-T cells showed the first signs of B-cell improvement only after six months [146]. In a retrospective study of 85 patients with DLBCL, CD4+ T-cell counts decreased from baseline and were persistently low one year after CAR-T cell treatment, with a median CD4+ cell count of 155 cells/µL [147]. In another cohort of 41 patients with DLBCL treated with CAR-T cells (axicabtagene ciloleucel), only 40% of patients had detectable CD19+ B lymphocytes after 12 months, and 50% of patients had a CD4+ cell count <200 cells/µL after 18 months following infusion. This delayed immune reconstitution in the majority of patients requires particular attention to supportive care measures [148]. A current research priority to improve infectious complications and outcomes in CAR-T cell therapy is a better stratification of patient risk for immunological dysfunction in order to implement prevention strategies.

## 6. Discussion

B-cell lymphomas are a heterogeneous group of non-Hodgkin lymphomas with varying degrees of response to immunochemotherapy. For many years, the anti-CD20 rituximab enjoyed a dominant position as part of combination therapy, resulting in the improvement of response rates and in overall survival. Despite successes in achieving remissions, a proportion of patients remain who are not cured with first-line therapy. A new repertoire of targeted therapies is available today, with which some patients can continue to be salvaged and sometimes put into a state of chronic disease. Targeted therapy is starting to drive a paradigmatic shift in how we think about treating lymphoproliferative disease. This shift has recently materialized as we explore new endpoints related to immunological processes, upon which both the evolution of lymphoma growth and infectious risk of the host depend [149]. B-cell lymphomas are characterized by dysregulation at multiple levels.

The growth and progression of lymphomas are supported by the silencing of the host immune system. The immunosuppressive networks involve changes in cellular microenvironment composition and distinct signaling pathways, each of them acting electively in the various histotypes [150]. These mechanisms used to circumvent or to manipulate the immune system to produce an appreciable effect on the host’s defense against infections, autoimmune disorders and secondary malignancies [151].

The most complex and characteristic destruction of immune functions is manifested in CLL. In contrast, the structure and functions of the immune system in some lymphomas are an understudied area. From a paradigmatic disease such as CLL, it will possible to acquire useful information to help the recovery of the immune system in other B-cells lymphomas [152]. 

For now, success in restoring the immune system has only been observed with ibrutinib in CLL, likely due to the longer follow-up period. Although treatment with new targeted therapies has profoundly improved outcomes and life expectancy, infectious risk still remains one of the main problems to be faced. Consequently, the immune failure underlying infectious risk before and after treatment remains a crucial issue.

The picture becomes more complex if we consider that targeted therapy is used in a patient population with a high median age, with multiple comorbidities, ineligible for autologous transplantation and with high-risk diseases. In addition, there is certain evidence that supports a role for age-associated changes in immune cell function that may impact targeted therapy results [153]. As a result, most lymphoma patients treated with targeted therapy are to be considered, in all respects, a special population.

Despite the encouraging data, the widespread introduction of immunomodulators and targeted therapy for the treatment of B-cell lymphoma into clinical practice needs to find an answer to many questions. What accounts for the immunosuppression associated with different lymphoma histotypes? Is it the entities themselves, their treatment, or both? How is infection risk influenced by sequential treatment with different types of chemotherapies, immune therapies and targeted therapies? How do new strategies affect infection risk and improve immune responses? What are the long-term effects of immune modulation? It is also important to establish which laboratory variables can predict infection risk and guide prophylaxis strategies (e.g., quantifiable changes in neutrophils, monocytes, T-cell subsets, natural killer cells, Igs). The roles of aging, fitness status and comorbidities in influencing infectious risk must also be explored. As this review shows, there are a number of stimulating new data on these issues that feed different streams of research. However, the issue of immunological health and the trajectory of reconstitution over time of the different immunological functions in patients treated for lymphoma still need to be addressed, at least for many settings. Finally, the ongoing COVID-19 pandemic makes the study of these key problems more urgent.

## 7. Conclusions

Immune impairment is the hallmark of lymphoproliferative diseases. The interactions of tumor cells with the players of the immune system not only facilitate the progression of the neoplasm, but also contribute to the worsening of overall health. The goals of new therapeutic advances are three-fold: to minimize the short- and long-term toxicity, to restore the immune system and to turn these diseases into curable malignancies. Drawing from studies that demonstrate both action against lymphoma and recovery of immune function, it is possible to show that targeted therapy can effectively restore immune homeostasis in the longer term. In future investigations, several strategies should be used, including prospective and registry studies, to identify immunological changes and risk factors for infections related to new treatments. It is to be hoped that the resulting knowledge will be increasingly integrated in future clinical decisions.

## Figures and Tables

**Figure 1 ijms-23-03368-f001:**
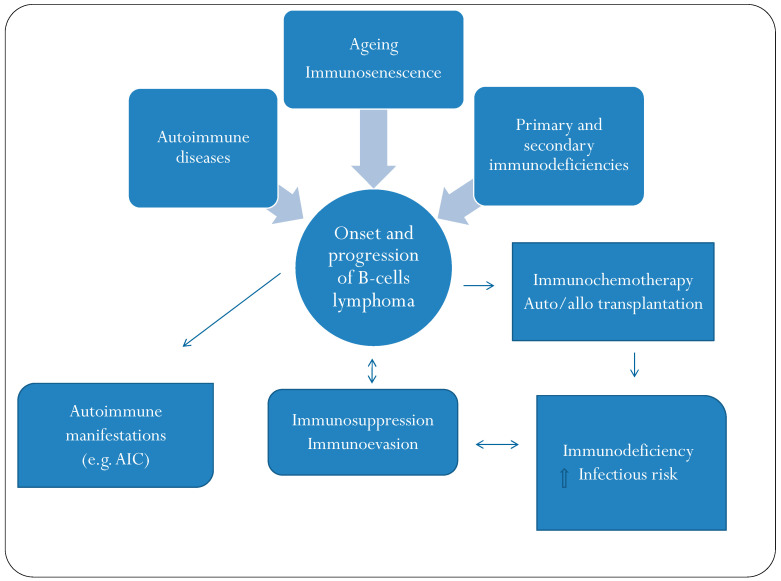
The dynamic relationship between the immune system and B-cell lymphoma: the mutual interactions between the lymphoma and the immune system are active through all stages of the natural history of the disease. AIC: autoimmune cytopenias.

**Table 1 ijms-23-03368-t001:** Selected studies on the effects of ibrutinib on the recovery of immune functions.

Study	Disease	Design	Patients, *n*.	Functional Effect
Long et al. [17]	CLL	Prospective	19	CD4^+^ and CD8^+^ expansion; decreased Treg/CD4^+^ T ratio
RESONATE RESONATE-2, Solman et al. [24]	Naïve and relapsed/refractory CLL	Prospective	55 50	Normalization of B cells, regulatory T cells, effector/memory CD4^+^ and CD8^+^ T cells, NK and MDSC counts
Solano de la Asuncion et al. [90]	CLL	Multicenter observational	23	CMV-specific T-cell expansion
Sun et al. [92]	Naïve and relapsed/refractory CLL	Prospective	84	Increase in serum IgA
Yin et al. [93]	CLL	Prospective	15	Normalization of T-cell numbers and T-cell-related cytokine levels; increase in T-cell repertoire diversity
AIM trial Ibrutinib + Venetoclax Davis et al. [104]	MCL	Prospective	24	Normalization of CD4^+^ and CD8^+^ effector and central memory T cells and natural killer cells

## Data Availability

Not applicable.

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
