# Peer review of "Effects of B-Cell Lymphoma on the Immune System and Immune Recovery after Treatment: The Paradigm of Targeted Therapy"

_ijms, 2022, doi:10.3390/ijms23063368_

Round 1
Reviewer 1 Report
The review by Salvatrice Mancuso and colleagues compiles very comprehensively literature to the topics of B-cell lymphomas and targeted therapies of immune functions. While the authors invested a great deal of work, the review is rather difficult to read even for researchers closer to the field. The two reasons for this are (i) the combination of effectively two topics – B cell lymphomas and targeted immune therapy – in one review and the unexplained usage of abbreviations and specialist terms.
As a review should capture the interest of a wider readership who is not yet so familiar with the field, I would like to suggest the following changes:
After a brief introduction to B-cell lymphomas illustrating the general clinical challenges, the information of section 2 could be intertwined with the section 3. Relevant information could be mentioned at the point where it is needed, while surplus information might be deleted. Hence, the review could start after the brief introduction with the current section 3 (section 2 would dissolve in section 3). That would shorten the text while putting the focus on the targeted therapies.
The second change refers to the specialist terms and abbreviations. Please write for an educated audience that is not familiar with the topic. Here a few examples of terms that would need explanation and/or context: L78 C1-C4, C3b; L88 NKG2D receptor; L90 T stem cell memory; L99 Tregs; L102 chronically activated phenotype; L284 hypogammaglobulinemia; L285 neutropenia, L289 reactivation of human HBV virus; L291 pn pneumonia; L293 CHOP vs R-CHOP; L380 cytomegalovirus (CMV) DNAemia; L413 Car-T cells; L415 immune checkpoint; L445 BTK inhibitor; L450 PI3K signalling; L472 BCL-2; L482 granulopoiesis; L509 CTLA; L510 PD1/PDL1 ligand 2; L529 Lenalidomide; L560 Lymphopenia.
Current the review assumes that the reader knows all of these concepts and proteins. This is however not the case with most readers.
Author Response
- “The review by Salvatrice Mancuso and colleagues compiles very comprehensively literature to the topics of B-cell lymphomas and targeted therapies of immune functions. While the authors invested a great deal of work, the review is rather difficult to read even for researchers closer to the field. The two reasons for this are (i) the combination of effectively two topics – B cell lymphomas and targeted immune therapy – in one review and the unexplained usage of abbreviations and specialist terms”
Reply: We thank the reviewer for the careful evaluation of the paper and for the observations she/he reported. We understand the complexity of reading the work that partly depends on the need to address the different aspects that concern the purpose of the review. In our opinion, the issue of the potential restoration of immunological functions with targeted therapy cannot be separated from the dissoul of the alterations of the immune system produced by lymphoma and chemotherapy treatments. The use of abbreviations is justified by the need to make the text flowing. In addition, technical terms are unfortunately not replaceable.
- “After a brief introduction to B-cell lymphomas illustrating the general clinical challenges, the information of section 2 could be intertwined with the section 3. Relevant information could be mentioned at the point where it is needed, while surplus information might be deleted. Hence, the review could start after the brief introduction with the current section 3 (section 2 would dissolve in section 3). That would shorten the text while putting the focus on the targeted therapies”.
Reply: On the possibility of dissolving part 2 with part 3 we think it is a difficult and counterproductive operation for the purpose of simplifying and lightening the text. B lymphomas are very heterogeneous and affect immune functions in different ways. A single chapter that talked about both the contents of part 2 and those of part 3 would risk being even more complex and difficult to read. On the other hand, the aim of the review is to address both issues. The first, that of the immunological alterations produced by different lymphomas, is little addressed by the current literature especially in the perspective of the effects of the treatments.
- “The second change refers to the specialist terms and abbreviations. Please write for an educated audience that is not familiar with the topic. Here a few examples of terms that would need explanation and/or context: L78 C1-C4, C3b; L88 NKG2D receptor; L90 T stem cell memory; L99 Tregs; L102 chronically activated phenotype; L284 hypogammaglobulinemia; L285 neutropenia, L289 reactivation of human HBV virus; L291 pn pneumonia; L293 CHOP vs R-CHOP; L380 cytomegalovirus (CMV) DNAemia; L413 Car-T cells; L415 immune checkpoint; L445 BTK inhibitor; L450 PI3K signalling; L472 BCL-2; L482 granulopoiesis; L509 CTLA; L510 PD1/PDL1 ligand 2; L529 Lenalidomide; L560 Lymphopenia”
Reply:
- L 78: as for C1-C4, C3b they are mentioned within a sentence where we talk about "complement components". In our opinion repeating that they are components of the complement would be redundant.
- L 88: NKG2D receptor: we specified the role of the receptor by adding the sentence “an activating receptor that plays a key role in the immune response against cancer”.
- L 90: T stem cell memory: the sentence explains that it is a sub-set of T lymphocytes. Repeating it would be redundant.
- L99 Tregs: two lines above the identity of the Tregs has been specified à Regulatory T cells.
- L102 chronically activated phenotype: we changed “chronically activated phenotype” with “ as a result of chronic activation”.
- L 284: we changed “hypogammaglobulinemia” with “reduced serum immunoglobulin levels”.
- L285: we changed “neutropenia” with “reduced neutrophil count”.
- L289 reactivation of human HBV virus: reactivation of hepatitis b virus is the reappearance or rise of hepatitis B virus DNA in the serum of patients with past or chronic HBV infection. It is a known phenomenon in the context of immunosuppression and we believe that the modality we have used to express it is abundantly used in the scientific literature and therefore easy to understand.
- L291 pn pneumonia: we have changed with Pneumocystis pneumonia (PCP). We have also changed Pneumocystis pneumonia with PCP.
- L293 CHOP vs R-CHOP: we have changed with “CHOP regimen (Cyclophoshamide, Hydroxydaunoribicin, Oncovin, Prednisone) versus R-CHOP (Rituximab plus CHOP).
- L380 cytomegalovirus (CMV) DNAemia; we changed with “CMV DNA in peripheral blood”.
- L413 Car-T cells: we had already specified the identity of CAR-T cells “Chimeric Antigen Receptor T cells”.
- L415 immune checkpoint: it refers to drugs that inhibit proteins called checkpoints, which are produced by cells of the immune system. It is the technical term widely used in scientific literature and therefore it cannot be changed or otherwise explained.
- L445 BTK: in L351 the enzyme BTK is described.
- L450 PI3K signaling: refers to a known pathway that promotes an intracellular response to extracellular signals. It would be excessive to explain in the text the activities promoted by this pathway.
- L472 BCL-2; in L 451 we have added the following sentence which best describes the BCL-2 protein : ”BCL-2 protein has the most prominent function of regulating the initiation of intrinsic pathway of apoptosis”.
- L482 granulopoiesis: we have changed with “production of neutrophils”.
- L509 CTLA; in L421 is described identity of CTLA (Cytotoxic T-lymphocyte Antigen).
- L510 PD1/PDL1 ligand 2: PDL1/PDL2 are the ligands of PD-1 receptor and together they form an inhibitor system that controls cell death. These technical terms cannot be modified and are widely reported in scientific publications.
- L529 Lenalidomide: lenalidomide is the name of the drug whose action is explained in the paragraph that concerns it.
- L560 Lymphopenia. We have changed “lymphopenia” with “lymphocytopenia”.
- “Current the review assumes that the reader knows all of these concepts and proteins. This is however not the case with most readers”
Reply: we understand the difficulty of reading this review by those who do not know the topics covered, which are both immunology and onco-hematology. However, considering the profile of the IJMS which is addressed to an audience of specialists, I believe that this paper is compatible with its aims and scope.

Reviewer 2 Report
In present review, authors attempt to shed light on immunological aspects of lymphoproliferative diseases and better ways of its targeting. I have several reservations; my comments are appended as below:
- Last part of abstract- its very confusing with its take home message. Authors should modify the same.
- While quoting studies with patients, authors should elaborate in details on no of patients included and statistical inference. For instance, reference 3,4., line 108- reference 20 This should be followed in complete manuscript.
- Authors should first describe the standard of care, observed prognosis and then move on targeted therapies.
- Figure 1- authors should include in brief mechanisms involved and elaborately discuss in text.
- Line 77-79, 137-139- describe mechanism.
- Line 82- reference 12- specify the study.
- Line 95-98- does it includes profile of inhibitory receptors on immune cells?
- Line 107- correct
- Line 220-221- authors rightly point out the limitations of PD-1/PD-L1 targeting immunotherapies. There is review discussing other cofounders. PMID: 33076303, authors may refer and describe.
- Table 1: authors should add a column with statistical inference (HR, P value)
- Line 228-230- reference. Authors should scan complete manuscript.
- Line 256- describe the type of chemo.
- Effects of targeted therapies and next section including CAR T cell therapy: authors should describe the off target effects, particularly observed in clinic and add a note on future directions.
- There should be ‘future directions’ section.
Author Response
According to Referee suggestion, we performed an English language editing.
We provide below, point by point, our revisions of the paper and our responses to the referee comments.
Referee 2:
- " Last part of abstract- its very confusing with its take home message. Authors should modify the same”
Reply: we thank the reviewer for the careful evaluation of the paper and for the observations she/he reported. We have changed “Aim of this review is to connect lymphoproliferative diseases, drugs and targets prioritizing immunologically relevant data with a view to adding elements of knowledge on the evolution of the immune system during and after treatment” with: “The aim of this review is to report relevant data on the evolution of the immune system during and after treatment with targeted therapy of B-cell lymphomas.
2." While quoting studies with patients, authors should elaborate in details on no of patients included and statistical inference. For instance, reference 3,4., line 108- reference 20 This should be followed in complete manuscript”
Reply: while agreeing with the reviewer's observation, it is difficult to report in detail numbers and statistics in the text because the possibility of treating lymphomas in the first line is now an acquisition amply demonstrated by experience as well as by numerous clinical trials. The same can be said for the numerous therapies that come into play in the second lines. As for reference 20, we have added the following sentence “These pseudoexhausted T cells may be a result of chronic stimulation by low-affinity self-antigens”.
3.“Authors should first describe the standard of care, observed prognosis and then move on targeted therapies”.
Reply: the description of the standard of care of the selected lymphomas, efficacy and prognosis of first-line treatments goes far beyond the objectives of the review. Implementing with this information would make the work longer. As a result it would conflict with the suggestions of Reviewer 1.
- “Figure 1- authors should include in brief mechanisms involved and elaborately discuss in text”
Reply: In the title of Fig.1 we introduced this sentence ”the mutual interactions between lymphoma and the immune system are active through all stages of the natural history of the disease”. We added in the text (L246): “Imbalance of immune system is closely relevant to the onset and progression of B-cell lymphomas. In addition, the same lymphoma impacts on the immune system to promote its progression. Finally, in synergy with immunochemotherapy, lymphoma causes a further imbalance of the immune defences”.
- “Line 77-79, 137-139- describe mechanism”
Reply: Line 77-79: we added a sentence: “The mechanism that determines complement deficiency is unknown. A genetic origin has been hypothesized, considering the finding of complement deficiency in healthy family members of patients with CLL”. With regard to lines 137-139 we believe that the mechanisms are described.
- “Line 82- reference 12- specify the study”
Reply: “A number of qualitative defects of neutrophil are been observed, including impaired phagocityc 80 killing of non opsonised bacteria and reduction of chemotaxis [12]”. In order not to lengthen the text further, we consider the sentence describing the alterations of neutrophils to be sufficient.
- “Line 95-98- does it includes profile of inhibitory receptors on immune cells?”
Reply: we can't understand the question.
- “Lines 107-correct”
Reply: The correction was made in “cells”.
- “Line 220-221- authors rightly point out the limitations of PD-1/PD-L1 targeting immunotherapies. There is review discussing other cofounders. PMID: 33076303, authors may refer and describe”
Reply: In L495, we added a sentence “the role of additional factors such as lifestyle, metabolic disorders and sociological factors in determining the response to this category of drugs must also be considered”. We have also added the bibliography reference.
- “Table 1: authors should add a column with statistical inference (HR, P value)”.
Reply: In general, there are several experiments reported in the individual studies, so it would be complicated and confusing to insert statistical indicators
- “Line 228-230- reference. Authors should scan complete manuscript”.
Reply: Bibliographic references are given
12.” Line 256- describe the type of chemo”.
Reply: we added a sentence: “containing e.g. high doses of Aracytin”.
- “Effects of targeted therapies and next section including CAR T cell therapy: authors should describe the off target effects, particularly observed in clinic and add a note on future directions”
Reply: We added this sentence :” Car-T cell-associated toxicities are related to severe supraphysiologic cytokine production and massive in-vivo T cell expansion”.
- “There should be ‘future directions’ section”.
Reply: we have integrated the concept of "future directions" at the end of the "conclusions" section. “For future directions” replaces “In this perspective”.

Round 2
Reviewer 2 Report
All my comments are well answered.